# Calcification of Various Bioprosthetic Materials in Rats: Is It Really Different?

**DOI:** 10.3390/ijms24087274

**Published:** 2023-04-14

**Authors:** Irina Y. Zhuravleva, Elena V. Karpova, Anna A. Dokuchaeva, Anatoly T. Titov, Tatiana P. Timchenko, Maria B. Vasilieva

**Affiliations:** 1E. Meshalkin National Medical Research Center of the RF Ministry of Health, 15 Rechkunovskaya St., 630055 Novosibirsk, Russia; a_dokuchaeva@meshalkin.ru (A.A.D.);; 2N. Vorozhtsov Novosibirsk Institute of Organic Chemistry SB RAS, 9 Lavrentiev Avenue, 630090 Novosibirsk, Russia; karpovae@nioch.nsc.ru; 3V. Sobolev Institute of Geology and Mineralogy SB RAS, 3 Academician Koptyug Avenue, 630090 Novosibirsk, Russia; titov@igm.nsc.ru

**Keywords:** bioprosthetic materials, cross-linking, calcification, collagen, elastin

## Abstract

The causes of heart valve bioprosthetic calcification are still not clear. In this paper, we compared the calcification in the porcine aorta (Ao) and the bovine jugular vein (Ve) walls, as well as the bovine pericardium (Pe). Biomaterials were crosslinked with glutaraldehyde (GA) and diepoxide (DE), after which they were implanted subcutaneously in young rats for 10, 20, and 30 days. Collagen, elastin, and fibrillin were visualized in non-implanted samples. Atomic absorption spectroscopy, histological methods, scanning electron microscopy, and Fourier-transform infrared spectroscopy were used to study the dynamics of calcification. By the 30th day, calcium accumulated most intensively in the collagen fibers of the GA-Pe. In elastin-rich materials, calcium deposits were associated with elastin fibers and localized differences in the walls of Ao and Ve. The DE-Pe did not calcify at all for 30 days. Alkaline phosphatase does not affect calcification since it was not found in the implant tissue. Fibrillin surrounds elastin fibers in the Ao and Ve, but its involvement in calcification is questionable. In the subcutaneous space of young rats, which are used to model the implants’ calcification, the content of phosphorus was five times higher than in aging animals. We hypothesize that the centers of calcium phosphate nucleation are the positively charged nitrogen of the pyridinium rings, which is the main one in fresh elastin and appears in collagen as a result of GA preservation. Nucleation can be significantly accelerated at high concentrations of phosphorus in biological fluids. The hypothesis needs further experimental confirmation.

## 1. Introduction

The first heart valve bioprosthesis made of a porcine aortic valve was implanted in humans more than fifty years ago. Since then, the problem of bioprosthetic calcification has been accompanying cardiovascular surgery. This problem has not yet been resolved, despite the large volume of new information on the physicochemical and biomolecular mechanisms of calcification, the progress achieved in materials science, as well as the new anti-calcification treatments in clinical practice [1].

It is well known that the younger the patient, the faster and more intensively the bioprosthetic calcification develops [1,2,3]. Thus, bioprostheses providing optimal hemodynamics and high quality of life while functioning normally are either not used in children and young patients, or their advantages are not realized due to rapidly developing dysfunction that requires reoperation. The cause of accelerated calcification in children and young patients is unknown. The primarily offered explanations, such as calcification dependence on “accelerated calcium metabolism” or “enhanced immunity” [2,3], appear too indistinct.

In recent years, it has become clear that not only GA-crosslinked xenogeneic bioprosthetic materials (e.g., porcine aortic wall, bovine jugular vein, and pericardium) can be calcified but also aortic and pulmonary allografts, tissue-engineered scaffold, polytetrafluoroethylene vascular prostheses [4,5,6,7,8]. Obviously, calcification occurs when the implant contains calcium binding sites and the recipient’s environment contains calcium and phosphate ions in an amount and ratio that facilitates the nucleation of insoluble calcium phosphates [9]. However, there is still no clear understanding of what exactly these sites are and what conditions of the recipient’s environment are necessary for nucleation to occur. Insufficient fundamental knowledge leads to making absolute any separate element of pathway and building concepts of calcification overcoming its principles. Such anti-calcification strategies that ignore other potential mineralization mechanisms do not allow researchers to comprehensively solve the problem.

For example, in the 1980s, the leading role of devitalized cell membrane phospholipids as the sites of calcium nucleation was the trending hot topic [10]. As a consequence, several anti-calcification methods using surfactants have been developed [11], but their clinical effectiveness has not been further proven. In addition, nobody currently can answer the question: why does GA-treated acellular collagen sponge calcify [12]?

Two decades ago, the discussion of recipient alkaline phosphatase (ALP) as a main calcification agent was popular, but the attempts to inhibit calcium deposition by immobilizing Al^3+^ and Fe^3+^ ions on implants were not very successful [13,14]. Thus far, the immune response to the Gal-proteins of porcine and bovine tissue as a cause of calcification has been vigorously discussed [10,15]. But then, how to explain in vitro biomaterials’ calcification in model calcium solutions [16] devoid of ALP and immune cells?

It was found empirically that post-fixation in alcohols has a mild anti-calcification effect [17]. The Lynx and XenoLogix technologies were developed using alcohol elimination of lipids [10]. However, the mechanism of alcohol/tissue interaction is not fully understood. We believe that this cannot be explained by the effect on lipids alone. Alcohols, in particular, isopropanol, significantly affect the conformational structure of the protein [17], increasing the content of random coils, β-sheets, and side chains, while reducing the level of antiparallel β-sheets and α-helices [18]. How this may be associated with bioprosthetic mineralization is still unclear.

The question of why the GA masking groups, which remain free after crosslinking, can participate in the nucleation of calcium phosphates [14] is also not disclosed; nevertheless, several methods of anti-calcification treatment of heart valve bioprostheses are based on this concept [13]. The desire to avoid the negative effects of GA led to the search for alternative crosslinking agents. Di- and polyepoxy compounds were the most popular at the end of the last century. Currently, methacrylates are being actively tested as alternative cross-linkers [19,20,21]. However, epoxides can effectively inhibit the calcification of collagenous materials (bovine and porcine pericardium and aortic valve leaflets) but not elastin-rich tissues (vein wall and, especially, the aortic tissue) [22,23,24,25]. There may be two reasons for this. Firstly, elastin, unlike collagen, cannot be crosslinked by GA or epoxides [26], which means that it contains its own calcium-binding sites independent of the cross-linker [27]. Secondly, elastin fibers contain many concomitant proteins [28], a significant part of which has Ca-binding properties—fibrillins, tenascins, vitronectin, etc. [29,30]. Perhaps one of these proteins promotes the calcification of elastin fibers.

In a word, “calcification, although mechanistically investigated for decades, remains a major impediment to the extended safety and effectiveness of bioprosthetic heart valves” [1]. The world scientific community has not received intelligible answers to the questions: what are the calcium nucleation sites in various biomaterials? Are their structures similar or different? What specific recipient factors contribute to the mineralization of the implants?

We hypothesized that calcium binding sites differ depending on the type of biomaterial: their chemical structure may be different in elastin-rich and collagen-rich xenogeneic materials, allografts, and tissue-engineered constructs. Moreover, various pathological processes, for example, Gal-dependent or bacterial inflammation, can form new sites, transforming the initially normal matrix structures. Of course, the most suitable objects to identify these sites are model proteins and peptides tested in vitro in calcifying solutions. On the other hand, living objects, including cases of subcutaneous implantation in small rodents, are able to answer the question about the chemical and morphological comparability of calcific deposits in different models and the nature of the accelerating or inhibitory effects on the calcification exerted by the recipient’s body with all its variety of enzyme systems, signaling molecules and cellular interactions.

The aim of this work was to quantitatively and qualitatively compare calcification features of collagen (bovine pericardium), and elastin-rich (porcine aorta and bovine jugular vein walls) bioprosthetic materials crosslinked with GA and DE in a rat model, as well as assess the potential role of fibrillin, ALP and recipient age in promoting mineralization.

## 2. Results

### 2.1. Fresh and Preserved Tissue Structure

The structures of fresh Pe, Ao, and Ve differed significantly. Pe contained only collagen fibers (blue fibers on Picro Mallory stained slides) with single cells (red) (Figure 1A). In the Ve wall, along with collagen, there were a large number of elastin fibers (yellow) and smooth muscle cells (red) (Figure 1B). There was very little collagen in the Ao wall, and the main part of the tissue was represented by elastin fibers and smooth muscle cells (SMCs) (Figure 1C). The tissue structure of all these materials did not visually change after crosslinking with both HAs and DEs.

### 2.2. Calcification Dynamics in Different Biomaterials

The calcium contents gradually increased in all implanted biomaterials (excluding DE-Pe) up to 50–85 mg/g in 30 days (Figure 1D). This amounts to a normal average by this date. DE-Pe did not accumulate calcium at all until 30 days. GA-Pe was the most calcified material (*p* < 0.05 compared with GA- and DE-Ao, GA-Ve, and DE-Pe), GA-Ao was the least calcified (*p* < 0.05 compared with GA- and DE-Pe; the difference with others was NS).

By the 10th day, the calcium accumulation was significantly lower (*p* < 0.05) in GA-Ao compared to other calcified materials. Both types of bioprosthetic veins did not show a significant difference on the 10th and 20th days, but the calcium content in GA-Ve increased significantly by the 30th day. In the group of DE-Ve samples, we noted the most variable results of tissue calcium content at every time point. Since the intensity of DE-Ve calcification can be associated mainly with elastin [21], these results may be related to varying elastin proportions in the Ve extracellular matrix. We found such differences in the Picro-Mallory staining of the specimens (Appendix A). The nature of these differences is not entirely clear. We assume that they may depend on the genetic characteristics of the breed, the age, and the diet of the animals.

The described patterns are indirectly confirmed by histological methods, which provide additional information about the predominant localization of the first deposits and the spread of the implant mineralization process (Figure 2 and Appendix A). Thus, the first calcium deposits in the venous material appeared in elastin fibers in the immediate vicinity of the subendothelial layer. In addition, devitalized SMCs are also a substrate for calcium deposits in GA veins. The initiation and progression of Ao-implant calcification occurred “from outside to inside,” starting in elastin fibers near the intimal and adventitial surfaces of the aortic wall. Then the mineralization process rushes inward, capturing more and more elastin fibers. In contrast, GA-Pe mineralization occurred “from inside to outside.” That is, the primary calcium phosphate deposits can be found in collagen fibers of the central layers of the tissue, then they gradually spread to the outer layers, which is accompanied by the increase of hydroxyapatite (HAP) crystals mass and destruction of the tissue in the central part. In DE-Pe implants stained by Von Kossa, we did not detect even trace amounts of calcium phosphates.

### 2.3. SEM and EDS Analysis Results

SEM and EDS analyses were performed on each sample, examining both the surface and the cross-section. The results showed that 10 days after implantation, only a few samples had calcium phosphate deposits on the surface. EDS showed that these formations could be represented by both rare foreign particles and calcium phosphate deposits of various compositions (Figure 3).

Foreign microparticles most often contained Fe and Cr, apparently adsorbed as a result of surgical instrumental manipulations or, less frequently, Si from dust. The vast majority of the particles were calcium phosphates with various Ca/P ratios. In some cases, this ratio showed a sharp predominance of Ca over P (spectrum 10 in Figure 3), and we attributed such particles to carbonate-apatite. However, in most cases, the Ca/P ratio in the surface formations ranged from 0.95–1.15 (Ao) to 1.25 (Ve) (Figure 4), which corresponds to immature apatite. The surface deposits in the Ao samples contained the least amount of Ca and P (Figure 4). It should be noted that Ao and Ve samples, regardless of the crosslinking method, differ less in phosphorus content than in calcium content. The significantly highest (*p* < 0.05) Ca content was observed in the GA-Pe, where the Ca/P ratio reached 1.6.

By the 20th day, a significant (*p* < 0.05) increase in the Ca content occurred in the Ao samples with a constant P content and, accordingly, an increase in the Ca/P ratio. In Ve implants, there are no significant differences in these parameters, while in GA-Pe calcifications, the concentration of Ca and P decreases in the absence of significant changes in Ca/P. This trend persists for up to 30 days. However, given that mineralization is not a surface process in all bioprosthetic materials implanted subcutaneously in rats (Figure 3), we believe that surface deposits are “random” formations located at the periphery of the process and cannot be the basis for making conclusions.

### 2.4. Micro- and Nano-Characterization of the Bioprosthetic Materials Mineralization

In fact, the mineralization process was somewhat different in the deep tissue layers (Figure 4, right column). Already on the 10th day, in all Ao and Ve samples, the Ca content in the deposits was 23–25 normalized %; in GA-Pe, it reached 31%, while the P content was approximately the same in all implants and amounted to 14–16 normalized %. The Ca/P ratio in the deep layers of Ao (1.24–1.25) and Ve (up to 1.35) testifies to the presence of more mature forms of calcium phosphates in the implants than on their surface. On the contrary, the content of Ca and P, as well as the Ca/P ratio, is lower in the cross-sectional samples of HA-P than on the surface. Further, up to the 20th day, all these implants showed an increase of Ca content (by 8–14 normalized %) with a very slight increase of P content (by 0–4 normalized %) and, accordingly, the increase of Ca/P ratio up to approximately 1.6. By day 30, the normalized Ca and P content, as well as the Ca/P ratio, were virtually unchanged, with a simultaneous increase in the mass concentration of Ca in the samples (Figure 1D). These dynamics allow us to hypothesize that in the first 10 days, calcium phosphate nucleation and formation of its “loose” deposits with an excess of phosphorus occurs. Then, up to the 20th day, these deposits not only grow in mass but also “rebuild” by saturating with Ca. Further development of mineralization is characterized only by an increase in the mass of crystals.

Light microscopy with von Kossa staining at ×400 was used to study the relationship between calcium deposits with various tissue structures (Appendix A). High-resolution scanning microscopy was used to characterize the morphology of the deposits themselves (Figure 5). To ensure that the calcified structures in the SEM images were correctly identified, a check was performed using the electron backscatter mode (Appendix A).

The data from both methods coincided well with each other as well as with the results of EDS, FTIR, and AAS. Thus, in both GA- and DE-Ao, calcium deposits were well visualized 10 days after implantation as rounded “growths” on elastin fibers. However, in the GA-Ao, calcium deposits were generally larger, and some of them had already begun to crystallize, and in the DE aorta, calcium phosphate was mostly represented by fine dust-like particles (Appendix A). SEM images clearly show that these deposits in the DE-aorta are 10–20 nm in size, while in GA-Ao—up to 0.5 μm. After 20 days, calcium phosphate in both types of Ao samples is mostly in the crystalline state (Appendix A), which is confirmed by the EDS data. By 30 days, the picture is the same in GA- and DE-samples: large confluent clusters of crystals on elastin fibers are present simultaneously with small crystals that continue to grow (Figure 5 and Appendix A).

GA- and DE-Ve showed a slightly different picture. Elastin was easily calcified in both types of implants, which could be observed by both light microscopy and SEM. However, massive calcification of smooth muscle cells and collagen could be detected in GA-Ve as early as day 10. By day 20, both extensive crystal growth and increased crystallinity occurred. After 30 days, GA and DE samples were virtually indistinguishable both in nano- and microscale deposits and in the amount of accumulated calcium (Figure 1D).

The GA-Pe, being a collagenous material, first demonstrated cardinal differences from the elastin-rich Ao and Ve counterparts; and second, only GA-Pe was calcified, while DE-Pe did not accumulate calcium at all (Figure 2 and Appendix A). Ten days after implantation, calcium deposits of various sizes, tightly adhered to collagen fibers, were clearly visualized in GA-Pe. Crystal growth and an increase in the Ca/P ratio (Figure 4) continued until the 20th day. By the 30th day, there was a significant fragmentation of the tissue due to its rupture by crystals. Some collagen fibers are totally replaced by calcium phosphates while maintaining their shape and D-periodicity (Figure 5).

Thus, it can be concluded that the Ca binding sites in elastin-rich materials are located predominantly in elastin. In collagen, binding sites appear only after treatment with GA.

### 2.5. Composition of Mineralized Deposits

The results of FTIR showed that 10, 20, and 30 days after subcutaneous implantation in rats, five of the tested biomaterials contained bioapatite (hydroxyapatite (*hap*) and carbonate apatite) and octacalcium phosphate (*ocp*, Ca_8_(HPO_4_)_2_(PO_4_)_6_ 5H_2_O) which is a metastable phase of biological apatites (Figure 6). The absorption bands of phosphate groups (1040, 605, and 566 cm^−1^) are clearly visible in FTIR difference spectra of both GA and DE Ao- and Ve-samples and GA-Pe already 10 days after implantation. It should be noted that the ν3 phosphate peak at 1030–1040 cm^−1^, characteristic of highly crystalline apatite [31], is predominant and fast-growing in all spectra. The accumulation of phosphates in the GA- and DE-Ao, and the GA-Pe proceeded gradually at an almost constant rate. In both vein samples, the formation of phosphate particles proceeded at a low rate between 10 and 20 days of implantation, but between 20 and 30 days, its amount increased several times, which generally agrees with the AAS data.

Along with the appearance of phosphate absorption bands in the spectra of all five materials, the appearance of amorphous calcium carbonate absorption bands is observed, indicating the formation of carbonate apatite. The DE-Pe does not accumulate phosphates or carbonates even after 30 days in rats. In the 1550–450 cm^−1^ region, the carbonate absorption bands include the 1430 cm^−1^ (asymmetric ν_3_ CO_3_) and 875 cm^−1^ (ν_2_ CO_3_) bands. The absorption band at 725 cm^−1^ (ν_4_ CO_3_) is absent in the spectra of the studied samples because it is characteristic of crystalline forms of calcium carbonate and is absent in amorphous forms [32].

The 875 cm^−1^ band also includes the stretching vibration of the P-OH of the HPO_4_^2−^ ion contained in *ocp* and apatite transitional forms [33,34]. The absorption bands of stretching vibrations ν_3_ of PO_4_ (1087 and 1038 cm^−1^) in the spectra of *ocp* and *hap* overlap; however, the presence of *ocp* can be seen from *in-plane* vibrations of OH groups in HPO_4_ (1206 cm^−1^) and stretching vibrations ν_3_ HPO_4_ (1134, 1118 cm^−1^) which give a broad shoulder in the spectra of differences of biomaterials implanted in the rat’s body during the first 10 days [35].

*Hap* and *ocp* can also be distinguished in the region 650–500 cm^−1^ of subtraction spectra. Appendix A shows a spectral profile of subtraction of the non-implanted GA-Pe spectra from spectra of implanted GA-Pe after 30 days of implantation.

The absorption bands in the spectral profile corresponding to hydroxyapatite are 633 cm^−1^ (OH^−^), 604 cm^−1^ (ν_4_ PO_4_^3−^), 579 cm^−1^ (ν_4_ PO_4_^3−^), 562 cm^−1^ (ν_4_ PO_4_^3−^). The absorption bands 579 cm^−1^ (ν_4_ HPO_4_^2−^) and 530 cm^−1^ (ν_4_ HPO_4_^2−^) correspond to the structure of *ocp*. The surface non-apatite hydrated calcium phosphate layer containing labile ionic particles is characterized by absorption bands at 619 cm^−1^ (ν_4_ PO_4_^3−^) and 546 cm^−1^ (ν_4_ HPO_4_^2−^) [36].

Thus, at the initial stage of calcification, there is an accumulation of phosphate and hydrogen phosphate ions in the calcifying structures. After 10 days of implantation, carbonates appeared, a larger amount of which was recorded in the GA-Pe and DE-Ao samples.

### 2.6. Potential Promoters of Calcification

#### 2.6.1. ALP Localization in Implants and Connective Tissue Capsules

Considering that ALP can penetrate into the implant only from the surrounding tissues, we visualized the “implant/recipient’s connective tissue” contact area. The images are presented in Figure 7.

Despite the fact that ALP appears in the capsule in an amount sufficient for identification already on the 5th day and remains there up to 30 days of observation, in none of the studied samples, we detected this enzyme directly in the implant tissue. It is important that we observed the maximum amount of alkaline phosphatase in the outer part of the capsule surrounding DE-Pe, in which there were no signs of calcification.

#### 2.6.2. Calcium-Binding Protein Fibrillin in Elastin Fibers of Aortic and Venous Walls

In Section 2.2 and our previous works [21,37], we have shown that the initial and predominant calcification of the Ao and Ve walls is observed in elastin fibers. Additionally, Ao and Ve showed a different onset and direction of calcification progress (Figure 2, Ao: inwards from the *laminae elasticae interna* and *externa*; Ve: near the internal elastic membrane towards the adventitia). We consider the calcium-binding role of elastin-associated proteins present in bioprosthetic materials to be an interesting but poorly studied issue. Since the composition of these proteins in arterial and venous walls is very diverse, the search for the proteins responsible for mineralization is a task of a separate study.

We hypothesized that the protein localized in the area of initial calcification could play the leading calcium-binding role.

Since there is a single study on the distribution of vitronectin (Vn) in arterial and venous walls [38], this question has not been studied with regard to other calcium-binding proteins. Therefore, we studied the localization of fibrillin, a potential candidate for calcification, in Ao and Ve walls.

It turned out that there is a large amount of fibrillin in the walls of Ao and Ve. It can be clearly seen (Figure 8) that it envelopes elastin fibers in a rather thick layer, especially in Ve. However, it is located uniformly throughout the thickness of the wall rather than in areas of initial calcification. Therefore, we consider the probability that fibrillin is the main Ca-binding protein of the Ao and Ve walls to be low.

#### 2.6.3. Ca and P Content in Rat Subcutaneous Facial Tissue

Calcification is usually simulated in young rats because this process is more intense in them. It is logical to assume that these differences are related to the different compositions of the recipient environment surrounding the implanted biomaterial. Since we are primarily interested in calcium phosphates, we investigated the Ca and P content in the subcutaneous fascia of rats. The results showed that upon desiccation, some of these elements are evenly distributed freely on the surface of the fascia, while the other part is electron-dense particles. We believe that these particles visualized by SEM (Figure 9) are the result of spontaneous evaporative crystallization of calcium phosphate from a metastable solution. Ca and P concentrations were found to be significantly dependent on age.

In young rats, P predominates over Ca both as a free element determined on the map (by a factor of 6) and in the particle composition (by a factor of 2). In aging rats, mapping revealed a 2-fold predominance of P over Ca, while surface particles, on the contrary, contained 2 times more Ca than phosphorus.

These differences between young and aging animals demonstrate that for the intensification of calcification, high phosphate concentration is more significant than calcium concentration.

## 3. Discussion

This is the first study in which the dynamics of calcification of bioprosthetic materials differing in ECM composition and preservation methods (GA- and non-GA) were studied using multiparametric assessment. Analyzing the results obtained, we hypothesized new mechanisms of bioprosthesis calcification.

A pronounced significant calcification of Ao and Ve, regardless of the crosslinking agent used, is observed as early as day 10 of subcutaneous implantation in young rats. Calcium deposits in Ao and Ve are always associated with elastin [9,39,40], are identical in micro- and nanostructure, and resemble primary calcifications in model elastin-like polypeptides [31]. We believe that the calcium phosphate nucleation sites in elastin are not related to fibrillin. We showed that fibrillin uniformly envelopes bundles of elastin diffusely across the cross-section of the samples, but primary calcium phosphate crystals can be found near the intimal surface in the vein wall and in the subintimal and subadventitial regions in the aortic wall, which does not correspond to fibrillin localization. We believe that vitronectin is a promising candidate as a calcification promoter because its distribution in arterial and venous walls corresponds to the foci of initial calcification [38]. K. Shina et al., who studied the role of Vn in ectopic calcium deposition, showed that the HX-domain of Vn can bind both soluble ionic calcium and crystalline HAP with high affinity and chemical specificity [29].

When Pe is crosslinked with DE, the tissue does not calcify throughout the experiment and even for a longer time [22,37]. In GA-Pe, the beginning of calcification is associated with collagen fibers in the central part of the specimen. Already on the 10th day, the calcium level and Ca/P ratio in GA-Pe is higher than in elastin-rich materials and reaches 1.44, which testifies to the rapid “maturation” of HAP in this material.

By day 20, the Ca/P ratio in all materials levels out, reaching about 1.6 (which almost corresponds to the mature HAP), and the further process is characterized mainly by an increase in the crystalline mass. By day 30, we observed that the Ca carbonate content increased markedly in all samples, indicating the relative stabilization of the mineral phase [31,41].

We are the first group to ask about the chemical mechanisms of the calcification of elastin and collagen bioprosthetic materials. What are the similarities and differences between these materials preserved by GA or non-GA methods?

It is known that the porcine aortic wall predominantly contains elastin and smooth muscle cells [42], and the bovine pericardium consists of type I collagen [43,44]. We further confirmed this in our work (Figure 1A–C). Literature data on the structure of the bovine jugular vein wall are very scarce, but, in our experience, the collagen/elastin ratio in it can vary depending on individual characteristics, age, breed of the animal, its diet, etc. (Figure 1B). Further, we proceeded from the fact that collagen does not calcify when it is fresh or not crosslinked with GA [12,22,23,24], whereas fresh elastin undergoes mineralization in the same way as it is crosslinked with an aldehyde or non-aldehyde cross-linkers [9,19,26]. Therefore, DE-Ve can be highly susceptible to calcification when elastin content is high (as in this work (Figure 1B)) or much less when collagen predominates [37] (Appendix A).

There are various hypotheses about the causes of the mineralization of soft tissues that are not normally calcified [45]. As for bioprostheses, we believe that the main cause of calcification is either the appearance of de novo HAP nucleation sites as a result of preservation or primary calcium-binding chemical structures in ECM proteins (as in elastin) or proteoglycans [46]. Given that the dynamics and morphological features of calcification of our materials differ little, we assume that their calcium nucleation sites are also similar. The identical chemical structure present in elastin (Equation (1)) and GA-linked collagen (Equation (2)) is positively charged nitrogen in pyridinium (Equations (1) and (2)). Back in the 1970s, it was discovered that pyridinium crosslinks predominate in GA-preserved collagen [47]. In fresh elastin, pyridinium is the core of desmosine, the main amino acid providing cross-linkage of elastin fibers and elastic properties of elastin [48]. We hypothesize that the initiation of HAP nucleation occurs as follows: in the presence of calcium and phosphate ions in the environment surrounding the implant, the positively charged pyridinium ring reacts with the phosphate anion, which, in turn, attaches Ca^2+^:



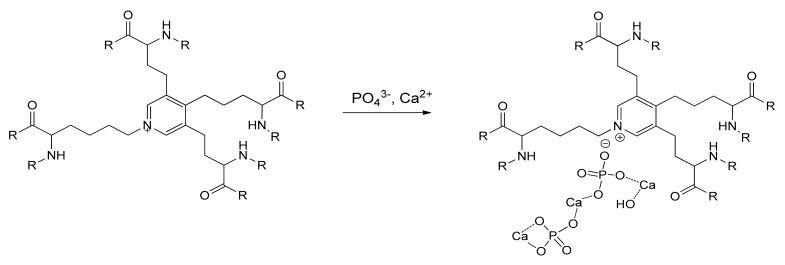

(1)

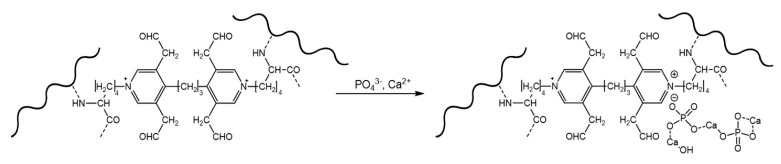

(2)

Indeed, phosphorus accumulation in the implants prevails in the early period of implantation (up to 10 days), and the Ca/P atomic ratio in the cross-sectional deposits indicates the predominance of non-crystalline calcium phosphates (Figure 4). After day 10, by contrast, calcium accumulation predominates, and Ca/P increases. Thus, the initial increase in local phosphate concentration plays a crucial role in HAP nucleation. The anti-calcification effect of bisphosphonates [49] bound noncovalently can indirectly confirm this. Their phosphonate groups may compete with the inorganic phosphate of biological fluids for positively charged nitrogen. This can lead to temporary blocking of mineralization.

Based on this hypothesis, we can expect that an increase in the level of phosphates in the biological fluids in contact with the implant may also contribute to its mineralization. We showed a significantly (6-fold) higher level of phosphate in the subcutaneous interfacial space of young rats, compared to aging animals, and no significant differences in Ca (Figure 9, Table 1). Since phosphate ions can play an initiating role in the initiation of mineralization, it is obvious that increasing the phosphate concentration in the implant environment will result in an acceleration of nucleation. We believe that this mechanism underlies the age-dependent intensity of bioprosthetic calcification. Empirically, researchers usually use young small rodents to study the calcification of xenogeneic materials in vivo [50]. Age-dependent accelerated calcification of GA bioprostheses implanted in young patients is also likely to be associated with an age-dependent increase in phosphate levels. It is known that from 0 to 90 years of life, the physiological level of phosphate in the blood decreases at least 2-fold [51], while the calcium level fluctuates in a very narrow range. Phosphate concentrations are very labile and depend on age, sex, pregnancy, kidney function, etc. [52].

ALP can affect calcification indirectly through the level of inorganic phosphate in the blood. As we have shown, ALP is secreted locally only by macrophages in the surrounding connective tissue in small amounts and does not penetrate into the implant. Therefore, we consider a significant effect of local ALP concentration to be unlikely, but systemic levels of this enzyme may be important. ALP can increase circulating inorganic phosphate levels by dephosphorylation of various biological molecules. ALP levels are highest before the age of 4 years and during puberty [53]. ALP levels decrease after 18 years of age and reach a plateau, which is accompanied by the stabilization of plasma phosphate concentrations.

We hypothesize that pediatric patients are more prone to bioprosthesis calcification precisely because of usually high levels of circulating inorganic phosphate and ALP. Adult patients with calcifying bioprosthesis dysfunction also have higher levels of this enzyme than patients without dysfunction [54].

This hypothesis concerns only specific types of calcification occurring in GA-treated collagenous and elastin-rich bioprosthetic materials. It needs further experimental development.

## 4. Materials and Methods

We used 25% glutaraldehyde (GA) (catalog No. 253857, Panreac Quimica SLU, Barcelona, Spain) and 97% ethylene glycol diglycidyl ether (DE) purchased from the N. Vorozhtsov Novosibirsk Institute of Organic Chemistry, SB RAS (Novosibirsk, Russian Federation) as a crosslinking agent.

### 4.1. Biomaterial Treatment

Fresh porcine aortic wall (Ao), bovine pericardium (Pe), and bovine jugular vein wall (Ve) were obtained from healthy animals immediately after slaughter and rinsed several times with 0.9% NaCl (Solopharm, St. Petersburg, Russia). Then half of each biomaterial was preserved at room temperature using 0.625% GA (0.1 M phosphate buffer, pH 7.4 for 21 days, with two solution changes on the 2nd and the 7th day) and another half with 5% DE (0.1 M phosphate buffer, pH 7.4 for 14 days, with one solution change on the 3rd day). Thus, 6 groups of bioprosthetic material were obtained. Ao, Ve, and Pe crosslinked with GA were stored in the same 0.625% GA solution; DE-preserved biomaterials were stored in the same 5% DE solution.

### 4.2. Subcutaneous Implantation of Biomaterials in Rats

All the experimental procedures were performed in accordance with EU Directive 2010/63/EU for animal experiments and approved by the Ethics Committee of E. Meshalkin National Medical Research Center.

Four-week old male Wistar rats (40–50 g, *n* = 90) were anesthetized with 50 mg/kg Zoletil (Virbac, Sante Animale, Libourne, France). Each animal was implanted with samples of all six types. Six incisions were made on the dorsal surface to prepare subdermal pouches. Each pouch was filled with one biomaterial sample (6 mm × 6 mm) and closed with one stitch. Twenty-eight samples of each biomaterial type were explanted on days 10, 20, and 30 and rinsed with 0.9% NaCl (Solopharm, St. Petersburg, Russia) (18 groups of biomaterials were studied in total). An additional 3 samples of each biomaterial type were taken out in 5 days for histochemical alkaline phosphatase (ALP) estimation.

### 4.3. Calcification Analysis

Ten explanted samples of each group were dried at 60 °C, weighed, and hydrolyzed in 14 M HNO_3_. Calcium quantification was done using a Thermo Solaar M6 atomic absorption spectrophotometer (Thermo Fisher Scientific, Waltham, MA, USA).

### 4.4. Histological, Histochemical, and Immunohistochemical Studies

All the samples for conventional histological studies were fixed in 10% neutral buffered formalin (Biovitrum, St. Petersburg, Russia) and then embedded in paraffin medium (Leica, Richmond, VA, USA) and cut into 6-μm thick slides. Three non-implanted samples of each biomaterial type were stained by Picro Mallory (Biovitrum, St. Petersburg, Russia) to visualize collagen, elastin, and SMC. Five explanted samples of each group were subjected to Von Kossa staining (Biovitrum, St. Petersburg, Russia), namely treatment with 5% AgNO_3_ for 30 min and then counterstained with eosin (pink staining) (Biovitrum, St. Petersburg, Russia) in order to identify the calcium phosphate.

Three samples of each group were explanted in blocks with surrounding tissue capsules and used for ALP histochemical visualization. Consecutive 5-6 µm cryosections of fresh samples (using cryostat Microm HM 550, Waldorf, Germany) were postfixed for 2 min in a mixture of absolute ethyl alcohol and acetone in a 1:1 ratio. Ready cryosections were stored in a freezer at −40 °C. The principle of the simultaneous azo coupling method was used for the histochemical localization of alkaline phosphatase (ALP) activity zones. On the day of staining, after rehydration for 4 min in 0.02 M PBS (pH 7.4), sections were incubated for 60 min at 4 °C with ALPase reaction medium, which contained 0,05MTris-HCl (pH 8.5), 5 mL; Fast Blue BB Salt hemi (zinc chloride) salt (Sigma-Aldrich, Milwaukee, WI, USA), 5 µg; DMSO, 500 µL; Naphthol AS-MX phosphate disodium salt (Sigma-Aldrich, Milwaukee, WI, USA), 2 µg. After, staining sections were rinsed in distilled water 3 times and covered by cover glasses with a mixture of distilled water:glycerol in a 10:1 ratio.

IHC fibrillin staining of paraffin-embedded samples was carried out with Ventana Benchmark XT immunohistostainer (Ventana Medical Systems S.A. Parc d’Innovation, Illkirch CEDEX, France) using Anti-Fibrillin 1 (ab231094) antibody (Abcam plc, Cambridge, UK) according to the manufacturer’s protocol.

The examination was performed using a light microscope (Axioskop 40, Carl Zeiss Microscopy GmbH, Gottingen, Germany).

### 4.5. Scanning Electron Microscopy (SEM) and Energy Dispersive Spectrometry (EDS) Analysis

#### 4.5.1. Explanted Samples Studies

Six samples of each group were straightened, fixed, and dried at room temperature under sterile conditions. Before the study, the samples were covered with a conductive carbon layer 10–15 nm thick on a Quorum Q150T ES (Quorum Technologies Ltd., Laughton, UK) device. In half of the samples, the internal surfaces were visualized (intimal for Ao and Be, serous for BP), while in the other half, the cross-sections were examined on a high-resolution scanning electron microscope MIRA 3 LMU with a Schottky cathode (Tescan Orsay Holding Brno - Kohoutovice, Czech Republic) equipped with an energy dispersive spectrometer (EDS) INCA Energy 450+ X-Max-80 (Oxford Instruments Nanoanalysis, Concord, MA, USA). The following conditions were selected: accelerating voltage of 20 kV, probe current of 1.5 nA, electron beam diameter of 8-9 nm, and spectra acquisition time of 20 s. The size of the X-ray generation zone in calcifications was about 2–3 μm. The smoothest and flattest areas of calcifications were selected for analysis. The analysis was performed in a spot mode or in the mode of scanning areas with an area of up to 10 × 10 μm^2^. Pure elements or simple synthetic and natural compounds were used as reference samples. Corrections for the matrix effect were calculated by the XPP method from the manufacturer’s software.

For a more detailed study of the samples, a Hitachi SU8200 (Hitachi, Tokyo, Japan) high-resolution electron microscope with a cold cathode was also used. The observation was carried out without preliminary deposition of a carbon coating at a voltage of 5–10 kV. In order to ensure that we correctly differentiate the calcified structures, verification was performed using backscattered electron mode (Appendix A).

#### 4.5.2. Rat Fascial Tissue Study

Experiments were performed on 3 aging (8 months old) rats with body weights 550–600 g and 3 young rats (4 weeks old) with body weights 40–50 g. Each animal was euthanized with Zoletil overdose; then, one longitudinal incision was made on the dorsal surface along the spine. The skin and the outer fascial layer at the edge of the incision were lifted with slight tension, and the fascial tissue located between the outer and muscular (inner) fascia was excised. This fascial tissue was presented as a thin film, which was subsequently spread on a hard horizontal surface covered with smooth aluminum foil. Such samples were dried at room temperature under sterile conditions and studied using SEM and EDS. The sample was fixed on the base of the sample holder (Appendix A).

SEM and EDS analysis of surface deposits as well as elemental mapping of tissues, was performed using a SU1000 FlexSEM II scanning electron microscope (Hitachi, Tokyo, Japan) equipped with the AzTec One EDX system (Oxford Instruments, Abingdon, UK). The samples were observed using a backscattered electron detector (BSE) at an electron beam energy of 15 keV and a pressure of 30 Pa (to avoid the charging phenomenon).

### 4.6. Fourier-Transform Infrared (FTIR) Spectroscopy

Four samples of Ao, Ve, and Pe preserved with GA and DE explanted from animals on days 10, 20, and 30 were washed with distilled water and dried at 37 °C to a constant weight. Fourier-transform infrared (FTIR) spectroscopy data were obtained using spectrometer Tensor 27 (Bruker, Bremen, Germany) using OPUS 6.5 (Bruker) software. The transmittance spectra (400–4000 cm^−1^) were collected as KBr pellets at a resolution of 4 cm^−1^ with 32 scans. A PIKE MIRacleTM ATR accessory equipped with a single reflection Zinc Selenide crystal was used for surface analysis at a resolution of 4 cm^−1^ with 64 scans in the range of 630–4000 cm^−1^. Baseline correction was made automatically by the rubberband method with 64 baseline points. The spectra were normalized to the 1655 cm^−1^ band (amide I) on the assumption that the content of amide groups in proteins was not affected by the preservation.

### 4.7. Statistical Analysis

Quantitative data were processed using Dell Statistica 13.0 (Dell Software Inc., Round Rock, TX, USA). Since the distributions of the majority of the groups were not normal, non-parametric statistics were used. Quantitative data are reported as medians and interquartile (25–75%) ranges (IQRs). The Mann-Whitney U-test was used to compare two groups, and the Kruskal-Wallis test was used to compare three or more groups. The level of significance was set to *p* < 0.05.

## 5. Conclusions

The results obtained in this work, together with literature data, allow us to build a logical concept of bioprosthetic calcification that occurs without accompanying pathological processes (nonspecific and immune inflammation, oxidative stress, etc.):

(1)Calcium phosphate nucleation sites are positively charged pyridinium rings present in the desmosine bonds of elastin fibers, regardless of the preservation method. In collagen, pyridinium rings are formed only after treatment with GA.(2)Pyridinium rings actively bind inorganic phosphate from the recipient’s blood, while the “pyridinium/phosphate” complex acquires a negative charge and the ability to bind cations in the blood, including calcium cations. After the formation of the nucleus, the process of self-assembly or self-organization begins.(3)Since inorganic phosphate ions (H_2_PO_4_^−^, HPO_4_^2−^, PO_4_^3−^) is a key link in the formation of the nucleus, the promoter factors of the recipient are all processes that lead to an increase in the concentration of phosphate ions in the blood (high ALP levels, impaired glomerular filtration of phosphorus, children’s age, etc.).

## Figures and Tables

**Figure 1 ijms-24-07274-f001:**
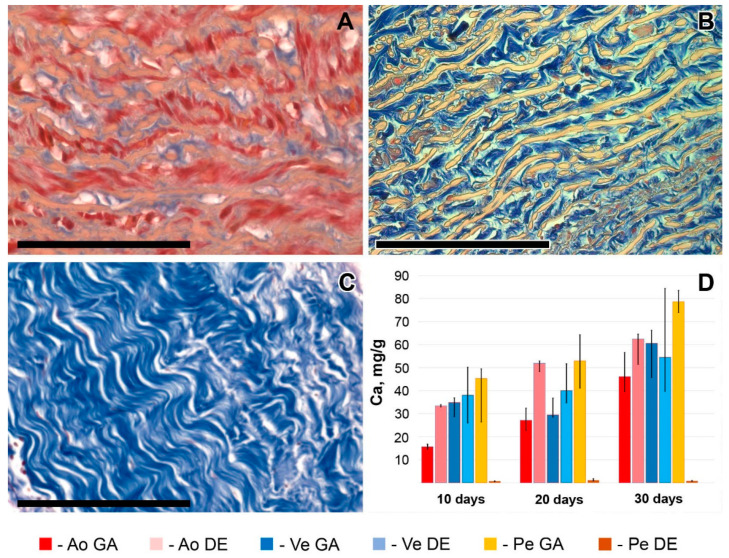
Different fresh biomaterial tissue structures: aortic wall (**A**), vein wall (**B**), and pericardium (**C**). Picro Mallory staining (**A**–**C**): collagen is blue, elastin is yellow, and smooth muscle cells are red. Scale bars are 100 μm. Calcium content dynamics in these biomaterials are preserved with GA and DE (**D**).

**Figure 2 ijms-24-07274-f002:**
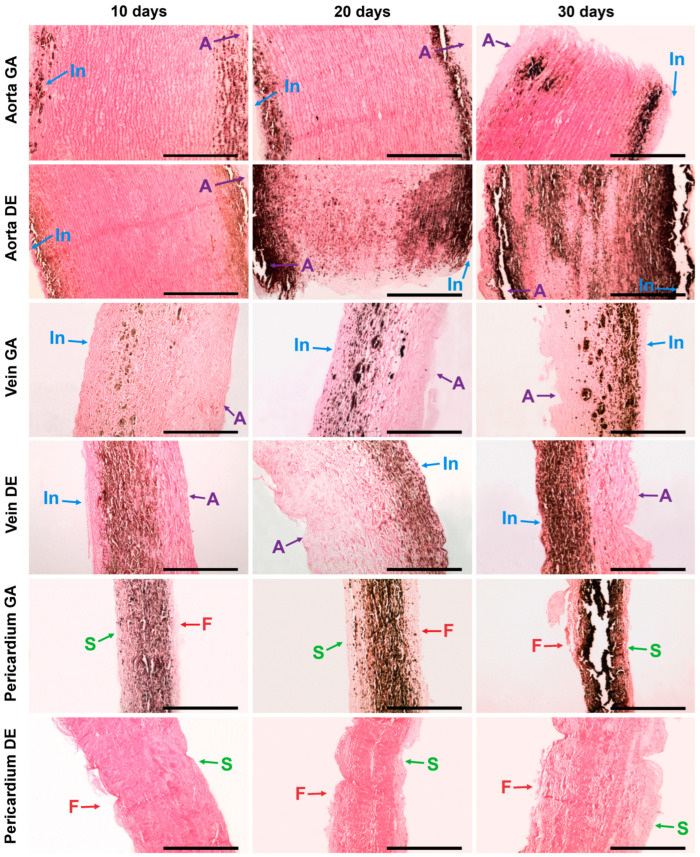
Calcium phosphate deposits in biomaterials 10, 20, and 30 days after implantation. Arrows show direction to surfaces: In—intimal; A—adventitial; S—serous; F—fibrous. No sign of calcification in DE-Pe. Scale bars 550 μm.

**Figure 3 ijms-24-07274-f003:**
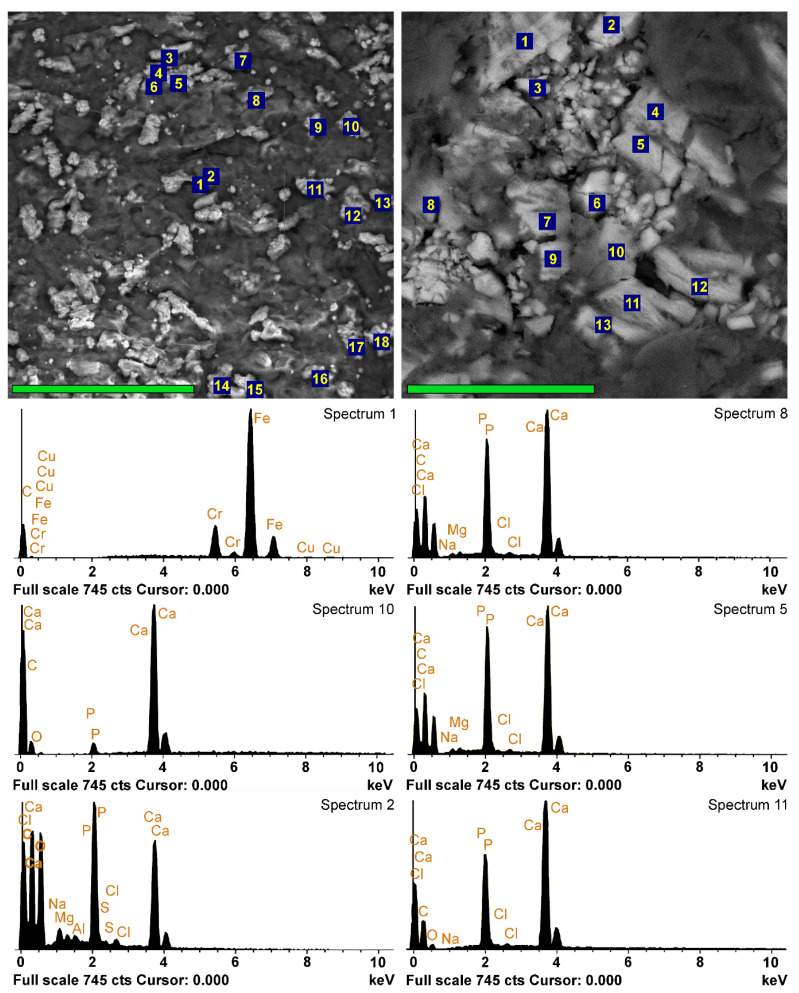
EDS analyses of the implants 10 days after implantation. SEM images in back-scattering electrons and EDS spectra of DE-Ve surface (left column, scale bar 80 μm) and cross-section (right column, scale bar 40 μm). Left spectrum 1 belongs to the external particle; spectrum 10 is hypothetically obtained from a carbonate-apatite deposit with a high content of CaCO_3_; all the other spectra belong to calcium phosphate with different Ca/P ratios.

**Figure 4 ijms-24-07274-f004:**
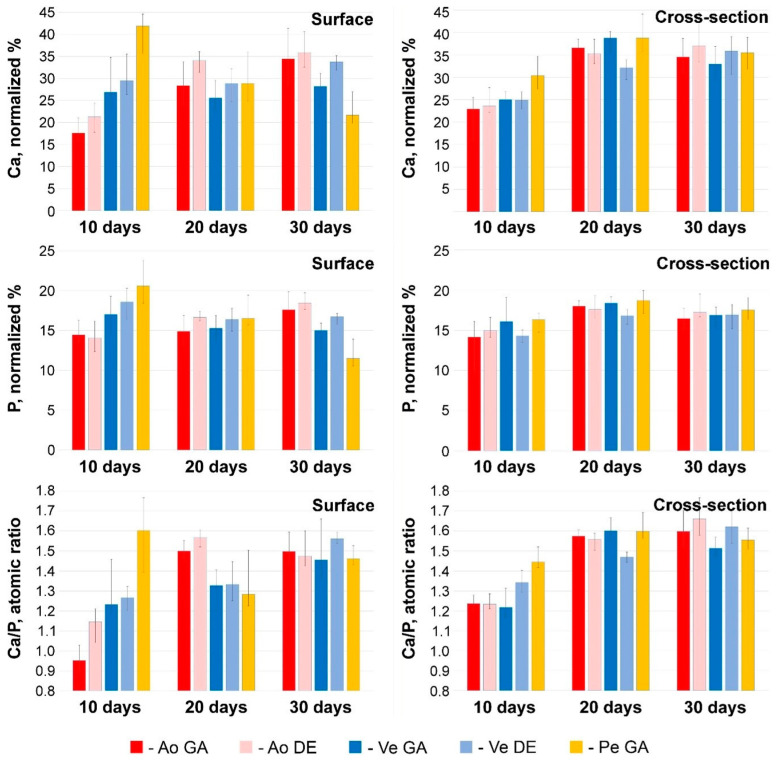
Results of EDS analysis at 10, 20, and 30 days after implantation. Ca and P content (normalized %) and Ca/P atomic ratios in electron-dense conglomerates located on the surface and in deep layers of tissue samples. Color labeling of biomaterials is similar to Figure 1.

**Figure 5 ijms-24-07274-f005:**
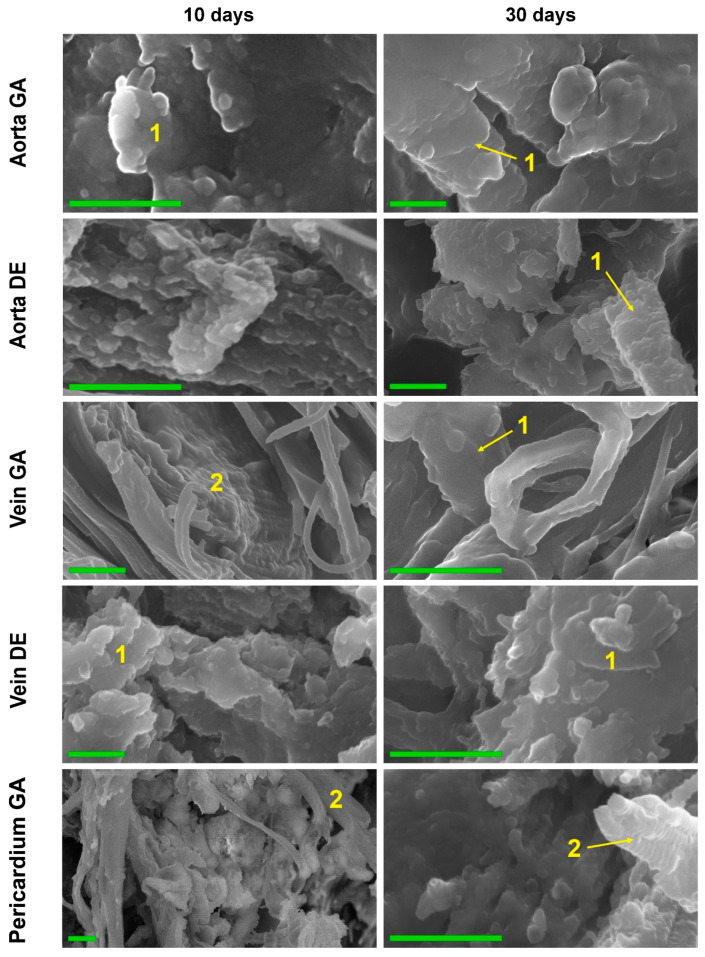
High-resolution SEM images of the biomaterials were explanted on post-implantation days 10 and 30. Elastin (1) and collagen (2) fibers calcification in 10 and 30 days. Collagen fibers completely retain the 3D structure, being totally replaced by calcium phosphates. The increase in the mass of HAP crystals and the formation of large HAP conglomerates after 30 days. Scale bars 500 nm.

**Figure 6 ijms-24-07274-f006:**
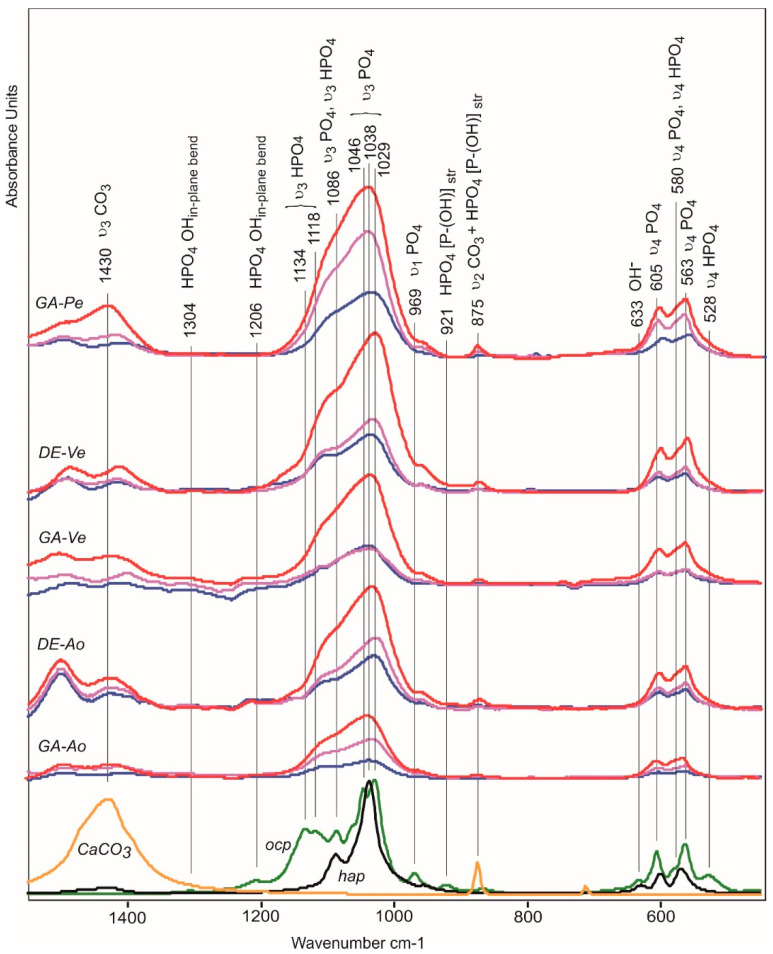
FTIR difference spectra as a result of subtraction of the non-implanted biomaterials spectra from spectra of implants after 10 (blue line), 20 (pink), and 30 (red) days of implantation. Bottom—spectra of calcium carbonate (yellow), octacalcium phosphate (*ocp*, green), and hydroxyapatite (*hap*, black).

**Figure 7 ijms-24-07274-f007:**
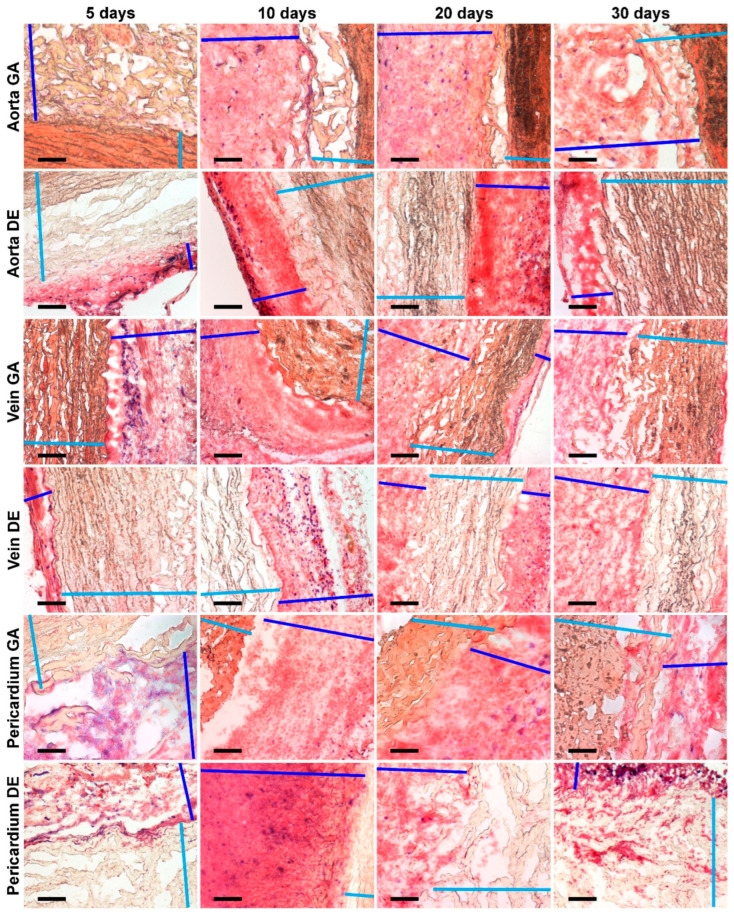
Histochemical visualization of ALP (deep blue spots) localized in connective tissue capsules but not in implants’ tissues. Light blue color marks implant tissues; deep blue lines show connective tissue capsules. Scale bars 200 μm.

**Figure 8 ijms-24-07274-f008:**
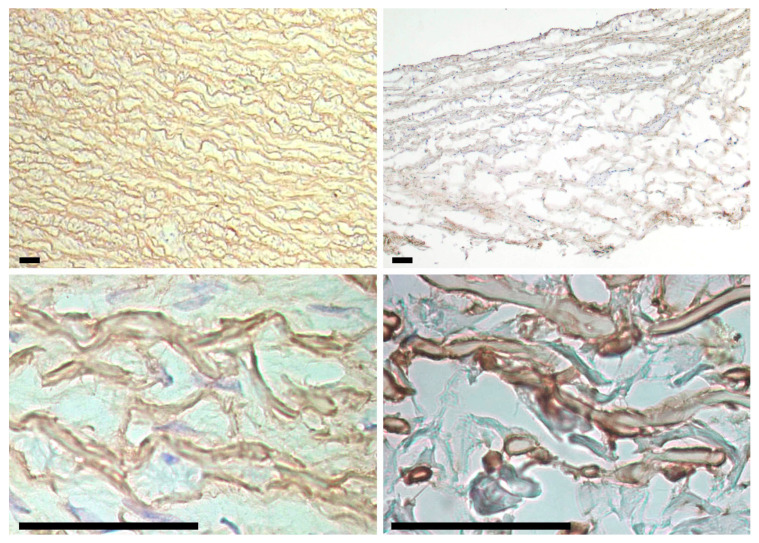
Fibrillin in porcine aortic (**left column**) and bovine jugular vein (**right column**) walls. Fibrillin is found in elastin fibers throughout the entire wall thickness (**top row**) and envelops the fibers outside (**bottom row**). Scale bars are 50 μm.

**Figure 9 ijms-24-07274-f009:**
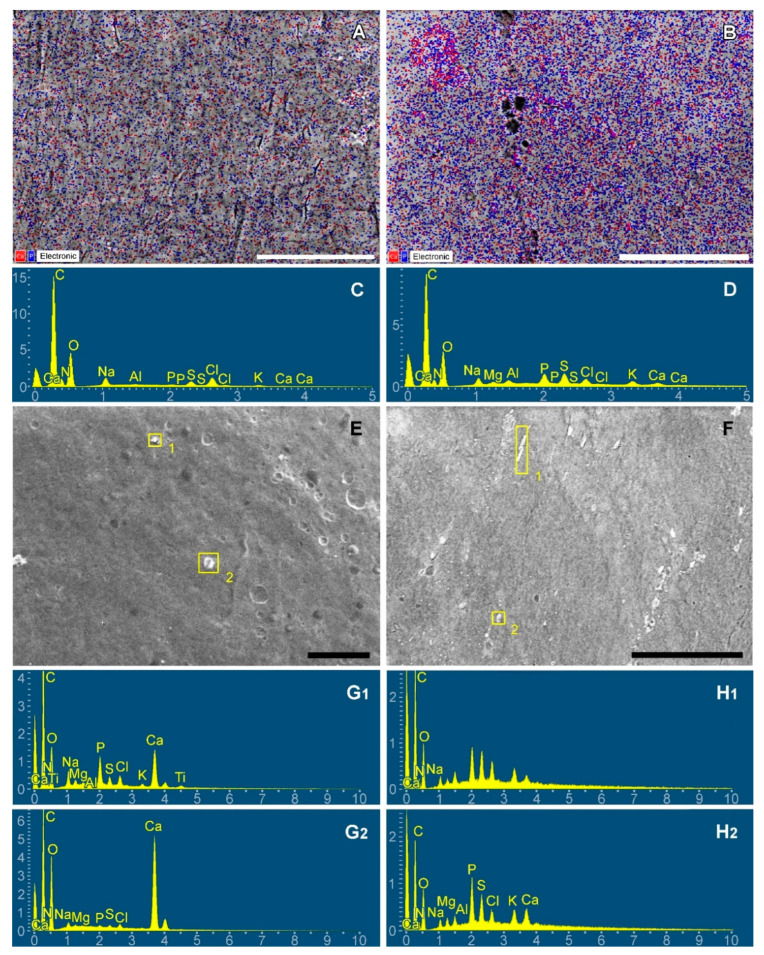
Ca (red) and P (blue) mapping of dry subcutaneous fascia excised from adult (**A**) and young (**B**) rats; EDS spectra of subsequent maps (**C**,**D**); calcium is visualized with red dots, phosphorus with blue ones. Electron-dense particles on the fascial surfaces: adult (**E**) and young (**F**) rats; spectra corresponding to particles 1 and 2 of each image (**G**,**H**). Scale bars 100 μm.

**Table 1 ijms-24-07274-t001:** Atomic Ca and P percentage, and Ca/P ratios in the subcutaneous fascia of young and adult rats. Data are presented as Me (IQR).

Elemental Map			Surface Particles		
Ca, Atomic %	P, Atomic %	Ca/P	Ca, Atomic %	P, Atomic %	Ca/P
Aging rats					
0.04 (0.03; 0.05)	0.06 (0.06; 0.08)	0.5 (0.5; 0.72)	0.41 (0.29; 0.61)	0.22 (0.16; 0.26)	2.1 (1.6; 3.8)
Young rats					
0.07 (0.03; 0.14)	0.35 (0.3; 0.55)	0.2 (0.14;0.25)	0.76 (0.51; 1.37)	1.65 (1.39; 1.92)	0.54 (0.34; 0.81)
P_aging-young_					
0.49	0.0004	0.0009	0.019	0.0003	0.0005

## Data Availability

Data is available within the article.

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
