# Peer review of "Calcification of Various Bioprosthetic Materials in Rats: Is It Really Different?"

_ijms, 2023, doi:10.3390/ijms24087274_

Round 1
Reviewer 1 Report
Material and Methods section is lacking
Cardiovascular tissue calcification and bioprosthetic valve calcification are two issue without a complete answer.Several studies have published, but we don't have prosthetic valves immune from this complication.
Author Response
Thank you for your work in reviewing our manuscript
- Material and Methods section is lacking
Materials and Methods section is not lacking. According to the rules of the International Journal of Molecular Science, this section is placed after the Discussion section. It is present in the manuscript under number 4.
- Cardiovascular tissue calcification and bioprosthetic valve calcification are two issue without a complete answer. Several studies have published, but we don't have prosthetic valves immune from this complication.
You are absolutely right, we agree that surgeons don’t have calcification-resistant bioprostheses. However, all approaches to solve this problem remind us of the Middle Ages, when, not knowing the cause of the pest, it was treated with prays. That is why we aspire to find the cause of calcification. Knowing the cause of the problem, it is easier to develop ways to solve it.
Reviewer 2 Report
The manuscript is devoted to the very substantial problem of medical science and surgery: what is the mechanism of calcification of xenogenic prostheses on the molecular level. This problem is actual at least last 30 years, but the definitive answer and solutions of this problem were not given. Calcification, although mechanistically investigated for decades, remains a major impediment to the extended safety and effectiveness of bioprosthetic heart valves. The authors, after the short review, formulate the main questions: what are the calcium nucleation sites in various biomaterials? Are their structures similar or different? What specific recipient factors contribute to the mineralization of the implants? The aim of the research to quantitatively and qualitatively compare calcification features of collagen in bovine pericardium and elastin-rich porcine aorta and bovine jugular vein walls bioprosthetic materials cross-linked with glutaraldehyde (GA) and diepoxide (DE) in rat model is of current interest.
The authors used a number of suitable instrumental methods to solve the problem: atomic absorption, histological methods, scanning electron microscopy, and Fourier-transform infrared spectroscopy. In the manuscript, many interesting novel results are described, between them the kinetics of Ca and P accumulation in different tissues, the morphology, composition, and localization of calcium phosphate deposits, as well as localization of potential promoters of calcification, etc.
The results obtained in this work, together with literature data, allow the authors to hypothesize the biochemical mechanism of bioprosthetic calcification in the absence of pathological processes. The main thesis is that calcium phosphate nucleation sites are positively charged pyridinium rings present in the desmosine bonds of elastin fibers. Pyridinium rings bind inorganic phosphate from the blade giving the ability to bind calcium cations. The conclusions seem to be realistic and novel.
The manuscript may be interesting and useful for the readers, so I recommend accepting it with minor revisions.
Only small remarks can be mentioned:
1. The abbreviation “SMCs” (line 119) appears before its decoding (line 147).
2. Arrows are absent in Fig.2.
3. Fig. S4 is described in the text of the manuscript after Fig. S4. The order of the Figures in the Supplementary Materials should be changed.
Author Response
We are grateful to the reviewer for the high evaluation of our work
Only small remarks can be mentioned:
- The abbreviation “SMCs” (line 119) appears before its decoding (line 147).
This mistake is corrected.
- Arrows are absent in Fig.2.
This mistake is corrected.
- S4 is described in the text of the manuscript after Fig. S4. The order of the Figures in the Supplementary Materials should be changed.
If we understand correctly, the reviewer means that Fig. S4 is described in the text earlier than Fig. S3. We checked it. Fig. S3 is first mentioned on line 135, and Fig. S4 on line 145. Therefore, we believe that the order of the Figures in the Supplementary Materials is correct.